# Technological and Strength Aspects of Layers Made of Different Powders Laminated on a Polymer Matrix Composite Substrate

**DOI:** 10.3390/molecules27041168

**Published:** 2022-02-09

**Authors:** Przemysław Golewski, Tomasz Sadowski

**Affiliations:** Department of Solid Mechanics, Faculty of Civil Engineering and Architecture, Lublin University of Technology, Nadbystrzycka 38, 20-618 Lublin, Poland; p.golewski@pollub.pl

**Keywords:** carbon fiber reinforced polymer, powders, finite element method, acoustic emission, three-point bending test

## Abstract

This study presents a description of the new technology for producing external or internal layers made of different powders mixed with epoxy resin, which can perform various functions as a protection against impact, erosion, or elevated temperatures as well as provide interlayers during the manufacturing of a ceramic protective barrier by air plasma spraying (APS) on the PMC substrate made of carbon–epoxy. Six types of powders (copper, quartz sand, Al_2_O_3_, aluminum, crystalline silica, and microballoon) were used to manufacture (120 °C) different kinds of protective layers (PLs), perfectly joined with the PMCs, in one single autoclave process. The two-layered specimens (2 × 25 × 110 mm) were subjected to a three-point bending (3-PB) displacement-controlled deformation process to determine the critical values of deformations at which the PLs can work safely without being cracked or delaminated. The tests were performed up to the final failure, observing various damage and cracking phenomena. Finally, the numerical simulations were carried out using the representative volume element (RVE) model of the most efforted central parts of the samples to determine the effect of powder grain diameter and resin content on the elastic properties and damage growth of the newly proposed multifunctional PLs. The stress concentrations and damage processes, including cracking and delamination, were analyzed in the whole two-layered system. The best result, in terms of strength during 3-PB testing, was achieved with the PL made of aluminum powder.

## 1. Introduction

The increase in the use of polymer-matrix composites (PMCs) is most noticeable in aviation. In the mid-1980s, the share of aluminum in aircraft construction was 74% and that for composites was 6%. In 2014, this proportion changed, and the use of aluminum dropped to 20% while there was an increase in the use of composites of up to 52%. However, further expansion of PMCs is limited due to their properties, such as low resistance to elevated temperatures and fire, low erosion resistance, and tendency to cracking and delamination under quasi-static or dynamic loading [1,2]. Such disadvantages can be eliminated to some extent by applying various multifunctional coatings on their surface. 

An extensive review of various coatings from biological to engineered coatings and techniques for their implementation was carried out in [3]. The following methods for the creation of protective layers (PLs) are widely used: atmospheric plasma spraying (APS), slurry method, reactive preparation, sol–gel, chemical vapor deposition (CVD), and microwave sintering. Other methods of deposition also exist, such as decorative layers [4] or improving tribological properties with the use of carbon nanotubes and graphene oxide [5]. Nanocellulose fiber-reinforced polymer coatings may also have potential applications [6,7,8]. The increased stiffness of the matrix can be attributed to the high stiffness of the fiber filler, which limits the movement of the epoxy polymer chains. Erosion properties and high-temperature resistance of the PMCs substrate can be improved by the application of the arc-spray method [9], particularly for material used in turbine engines that suffer degradation by small particles drifting by the air. In laboratory tests, samples with and without coating were grit blasted with 28 mesh corundum powder and 0.2 MPa compressed air. The mass loss of the coated PMC in erosion testing was half that of the uncoated one. The results of testing the erosion effect both at room temperature and at 250 °C for PMCs samples with a coating made by thermal spraying are presented in [10]. 

In the study carried out by [10], two types of PLs were manufactured using HVOF: (1) by flame spray and (2) by wire arc HVOF. The relatively low particle velocity during flame spray caused insufficient deformation and diffusion, leading to the top-coat layer being irregular and had low cohesion. Moreover, the erosion at 250 °C was twice as high as at room temperature. 

The application of the cold spray method for the creation of PLs in which high particle velocities are used was investigated in [11]. Two types of Al and Al/Cu layers were fabricated on carbon composite with a PEEK matrix. Both layers were homogeneous and dense with porosity equal to 1.1%. This cold spray method can be applied to composites with a thermoplastic matrix. However, one can point out the use of the cold spray for composites with a chemically curable matrix, such as epoxy, does not give positive results, i.e., only the erosion of the surface occurs. 

In [12,13,14,15,16,17], different types of coatings that improve the erosion properties of PMCs and allow them to operate at elevated temperatures or protect them from flame effects were investigated. Such coatings are called TBCs (thermal barriers coatings) [18,19] and are commonly used to protect hot parts of turbine engines [20,21]. Multi-layered TBCs were manufactured on the surface of the composite with thermoplastic BMI matrix [16] where the triple-layered system with the top-coat was made of yttria stabilized zirconia (YSZ).

Unfortunately, some of these methods, such as sol–gel, nanopaper, or ceramic fiber mat, do not protect the substrate from erosion despite excellent thermal insulation. 

To date, there is no work in the literature where a TBC system with metallic bond-coat MCrAlY and top-coat YSZ, made by hot methods, such as APS or HVOF, was fabricated on a PMCs substrate with an epoxy matrix. The solution to the problem is the application of an additional interlayer, for which strength and adhesion to the PMC substrate will determine the ability to produce a multilayer and effective TBC system that is durable during operation. 

In this study, a new manufacturing methodology by creating additional PL using six different types of metal and oxide powders with different properties, including copper, quartz sand, Al_2_O_3_, aluminum, crystalline silica, and microballoons, which were infiltrated by epoxy migrating from the PMC prepreg during one single autoclave process, is proposed. 

The aim of this study was to determine the strength and mechanical behavior of such complex layered structures; a 3-point bending (3-PB) displacement-controlled deformation process with acoustic emission was carried out. 

The aim of this study was also to determine the effect of grain size on the layer properties by using local numerical model taking into account a small fragment of the two-structure called representative volume elements (RVE) [22,23,24,25,26,27,28,29]. The numerical analysis was focused on the description of the damaging process in the selected RVE from the central part of the sample.

## 2. Materials and Methods

### 2.1. Materials and Their Characterization Methods

Kordcarbon prepreg (Fiberpreg GmbH, Neu-Ulm, Germany) with twill carbon fabric and an epoxy matrix was used to fabricate the PMC substrate for further deposition of six different PLs. Material data for the Kordcarbon prepreg used are available from [30], and from [31] for the epoxy matrix. The mechanical properties collected of the two components used for the creation of the protective system are shown in Table 1 and Table 2.

The PLs placed on the PMC substrate were created during an autoclaving process by a mixture of epoxy resin with the following metal or oxide powders (PolyCore, Świdnik, Poland): (a) copper, (b) quartz sand, (c) Al_2_O_3_, (d) aluminum, (e) crystalline silica, and (f) microballoon. The microballoon mainly consisted of SiO_2_ (76.81 wt%); the other compounds were CaO (12.98%), Na_2_O (5.27%), CuO (3.56%), and Al_2_O_3_ (1.38%). Figure 1 summarizes the SEM microscope images for each powder material, keeping the same magnification.

Each powder is characterized by both a different shape and grain size. Grain size measurements were obtained using SEM and are shown in Figure 2.

Both crystalline silica and microballoon are in the form of microspheres, thus making their specific gravity low. The wall thicknesses are 2.5–4 μm and 0.2–0.8 μm, respectively. Such low wall thickness in the case of the microballoon means that its structure can be easily damaged as can be seen in Figure 1f, where the vast majority of the spheres are cracked. 

For manufacturing of the PMC substrates, 8 layers of Kordcarbon prepreg were applied. After the layers were arranged and consolidated, a 1 mm thick frame was applied as a template to produce a uniform layer of powder. The powder was put inside the template and the excess powder was removed as shown in Figure 3a. After removing the template, a uniform thickness layer was created (Figure 3b) and subjected to the vacuum bag technology. The entire package was cured in an autoclave at 120 °C according to the prepreg manufacturer's guidelines. The procedure was repeated for six types of powders. 

After the curing process in the autoclave, all samples of dimensions 25 × 110 mm were cut using a CNC plotter. For experimental testing, three specimens per type of powder were prepared. Microscopic observations of the internal structure of the two-layered composites were created using the Keyence VHX-7000 digital microscope (Figure 4). The thickness of the PMC substrate layer was the same in almost every case and equal to 1.47–1.55 mm. The exception was the samples with the microballoon layer with the PMC substrate thickness of about 1.74 mm. Despite using the same bulk thickness of the powders, significant differences were obtained in their thickness after the curing process. This results from the different sizes of grains and their shapes. The space between the powder grains is filled with the resin coming from the Kordcarbon prepreg. When using this type of technique on a larger industrial scale, it is necessary to ensure that there is an adequate excess matrix of the prepreg that would fill the mound of grains in the powder layer. Microscopic observations show that the powder layers are perfectly joined to the PMC substrate, without visible discontinuities, pores, or voids. The powder perfectly fills the irregular surface of the prepreg, which is undulating due to the presence of weaves in the fabric.

### 2.2. Experimental and Numerical Methods

A displacement-controlled 3-PB deformation process (Figure 5) was performed using the MTS 100 kN testing machine, keeping the distance between the supports equal to 90 mm. The bending was carried out quasi-statically with the speed of the testing machine transducer 1 mm/min up to the failure of the samples. During the test, the PL part of the specimens was placed in the bottom tensile zone. To record the onset of damage and its further growth in the specimen, a piezoelectric sensor and the National Instruments data acquisition system were used to record the acoustic emission (AE) signals. The processing of the results was carried out by Diadem 2019 software (National Instruments, Austin, TX, USA).

Three representative RVE models were created in the two steps:
In the first stage, geometric CAD models were created in SolidWorks 2014 software (Dassault Systèmes, Vélizy-Villacoublay, France) for the PLs with the epoxy matrix. The external dimensions of this central fragment (RVE) in each case were the same and equal to 1 × 1 × 10 mm (Figure 6a,b). The models differed in the number and size of powder grains:
Model 1 contained 288 spherical grains with a diameter of 0.25 mm (Figure 6c);Model 2 contained 672 grains with a diameter of 0.19 mm (Figure 6d);Model 3 contained 1300 grains with a diameter of 0.15 mm (Figure 6e).In the second step, the geometric models of the PLs were imported to Abaqus software (Dassault Systèmes, Vélizy-Villacoublay, France) and the two-layered structure was completed by adding the PMC substrate part to the assembly (Figure 6f).

The structured grids were used for the creation of the finite element mesh. A connection between the PL part and the shell part (substrate) was created by using tie bonds. This type of constraint is justified by the lack of delamination between the PCM substrate and the PLs during experimental 3-PB testing. The damage growth in the epoxy matrix was assumed to be described as brittle microcracking. 

Due to the small size of the powder grains in the PLs, the finite element mesh should be dense. Therefore, the global element size for the epoxy matrix model was 0.019 mm, while it was 0.011 mm for the powder grain model. Such dense mesh is not required for the PMC substrate part, where the shell elements with a global dimension of 0.1 mm were enough. The PMC substrate layer was modeled using continuum shell (SC8R) elements in order to describe the damage process with the application of the Tsai–Hill hypothesis for composites. 

The total number of elements of the whole two-layered model ranged from approximately 3.2 to 4.1 million elements, depending on the number of grains in the PL.

The above degradation model also forces the explicit code in Abaqus. Due to a large number of finite elements in the description of the analyzed microcracking process it was necessary to scale the mass in order to be recalculated at each time increment in a relatively short time (few days). To keep the same conditions for all models, a target time increment = 5e^−6^ s was assumed.

The boundary and loading conditions applied for microcracking analysis of the 3D RVE models are presented in Figure 6g. The models were subjected to bending by the load uniformly distributed over the top surface of the PMC composite, with intensity increasing up to 10 MPa. Loading condition results from the experiments are shown in Figure 5, where the load is transmitted by roller. The boundary conditions were assigned to the side faces using reference points (RP1 and RP2) bounded by coupling ties to the side faces. For RP1, all three displacements were locked, while for the RP2, only two of them were locked with free displacement along the "x" direction.

## 3. Experimental and Numerical Results 

### 3.1. Experimental Results

Figure 7 presents the failure modes at 3-PB tests for six two-layered specimens with the PLs different powders. Different shapes of the fractured samples can be observed; however, there is no such significant delamination process of the layer as was presented in [19], where also a single autoclave process was used for joining the ceramic mat to the PMCs substrate. This is a great advantage of the proposed technology, where the powder can easily penetrate between the weaves of the fabric and also between the fibers under air pressure in the autoclave. The adhesion of the powder to the substrate is so high that, for example, in the case of the Al_2_O_3_ layer (Figure 7c), delamination in the substrate is visible, while the PL still adheres to the PMC layer. It is necessary to note that no plastic behavior was observed and crack propagated only in a brittle manner in the two-layered composites. 

It is notable the aluminum layer (Figure 7d) was found to be an exception, where the PL holds the PMC substrate in its highly curved shape, even at final failure. Delamination of the PL was absent and the damage of the substrate was not extensive. 

The important tool used in the testing of the cracking process of the two-layered system is the AE method, which was firstly used in the 1970s [32]. This technique is applied in different loading states, such as flexure [33], tensile [34], or double cantilever tests [35]. The idea of AE experiments is to find levels of signals amplitude (dB), energy, or emission frequency corresponding to the phenomena, such as fiber breakage, matrix cracking, interfacial debonding, delamination, fiber pull-out, and fiber slippage. The various descriptors of the AE method for application to PMCs are discussed in detail in [36].

In the case of the two-layered composite systems, the deterioration process is complicated as it is necessary to include three processes:
Matrix cracking in the PL;Delamination of the PL from the PMC substrate;Degradation of the PMC layers.

Figure 8, Figure 9, Figure 10, Figure 11, Figure 12 and Figure 13 present two plots: (1) bending moment-displacement correlations and (2) AE-displacement functions for all samples. The peak amplitude (*A*) can be expressed in dB as:(1)$$A=20\log\left(\frac{U_{\max}}{U_{ref}}\right)$$
where*U*_max_—represents the largest voltage peak in the recorded acoustic waveform.*U_ref_*—reference voltage is set at the pre-amplifier.

Figure 8 shows the first batch of results for three samples with the PL made of the copper powder. The first microcrack detected by the AE appears for the bending moment in the range 4.6–4.8 Nm and a displacement of 4.75–5.05 mm. The obtained results are very coincidental. The final failure of the central cross-section is indicated by both a slight decrease in force visible in the graph and an AE peak of 71 dB. 

**Figure 8 molecules-27-01168-f008:**
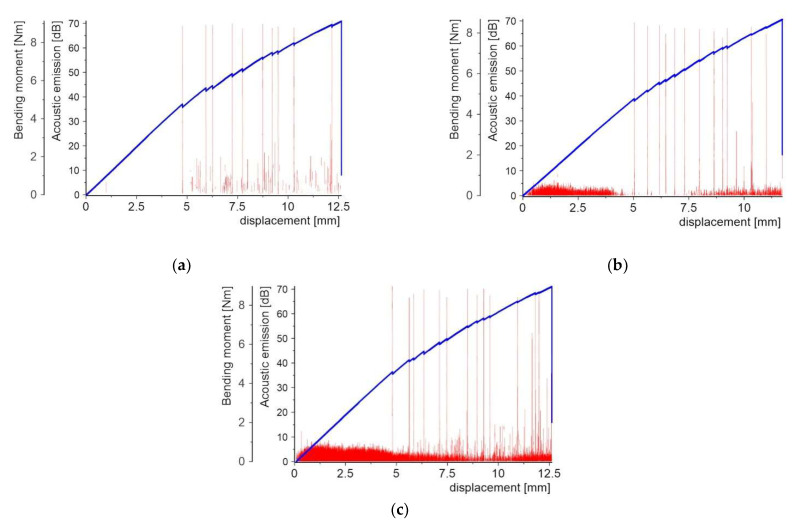
Bending moment and AE vs. displacement diagrams for 3 specimens with a PL made of copper powder: (**a**) sample 1_1; (**b**) sample 1_2; and (**c**) sample 1_3.

**Figure 9 molecules-27-01168-f009:**
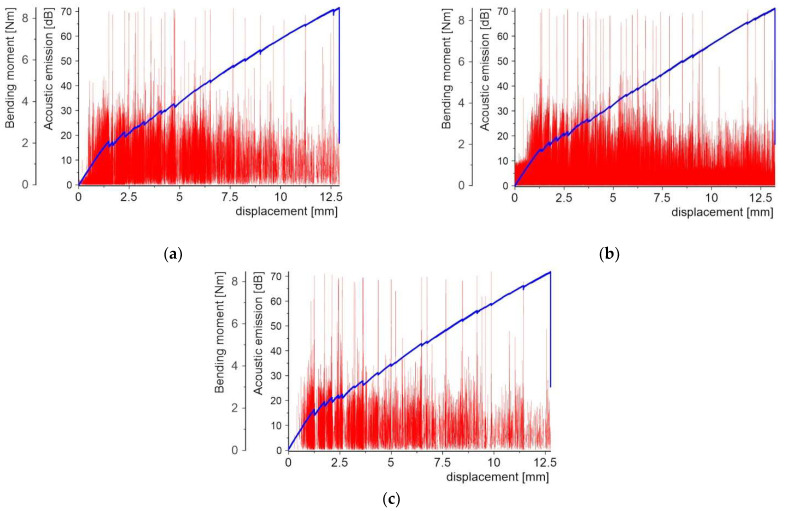
Bending moment and AE vs. displacement diagrams for 3 specimens with the quartz sand PL: (**a**) sample 2_1; (**b**) sample 2_2; and (**c**) sample 2_3.

**Figure 10 molecules-27-01168-f010:**
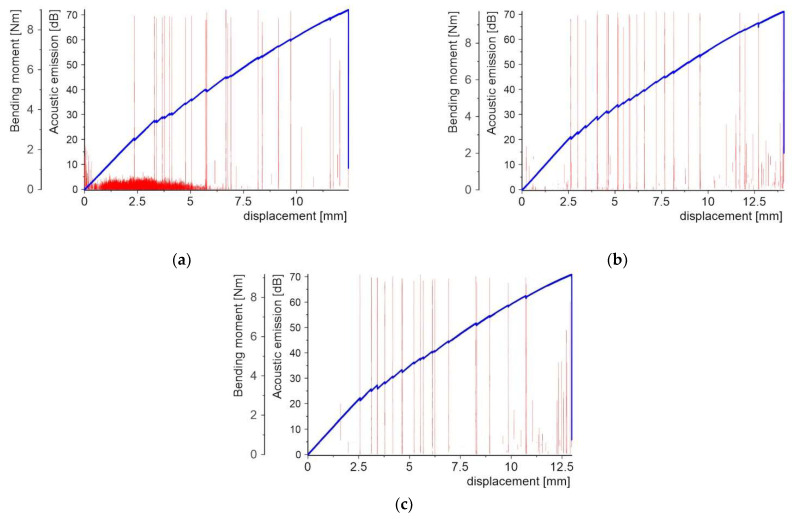
Bending moment and AE vs. displacement plots for samples with Al2O3 PL: (**a**) sample 3_1; (**b**) sample 3_2; and (**c**) sample 3_3.

**Figure 11 molecules-27-01168-f011:**
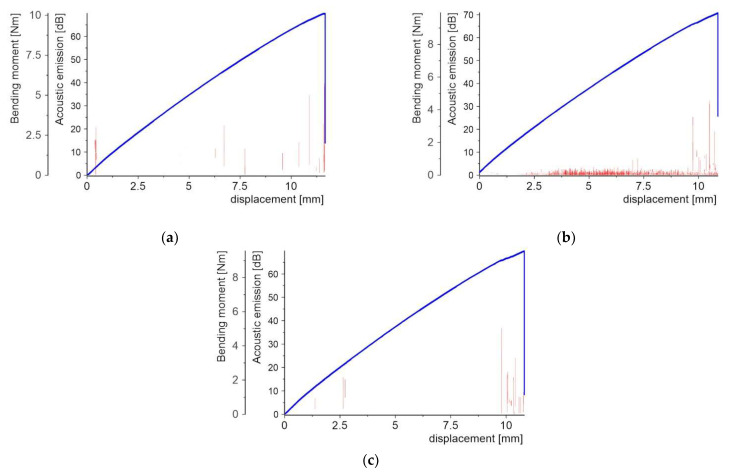
Bending moment and AE vs. displacement diagrams for specimens with aluminum powder PL: (**a**) sample 4_1; (**b**) sample 4_2; and (**c**) sample 4_3.

**Figure 12 molecules-27-01168-f012:**
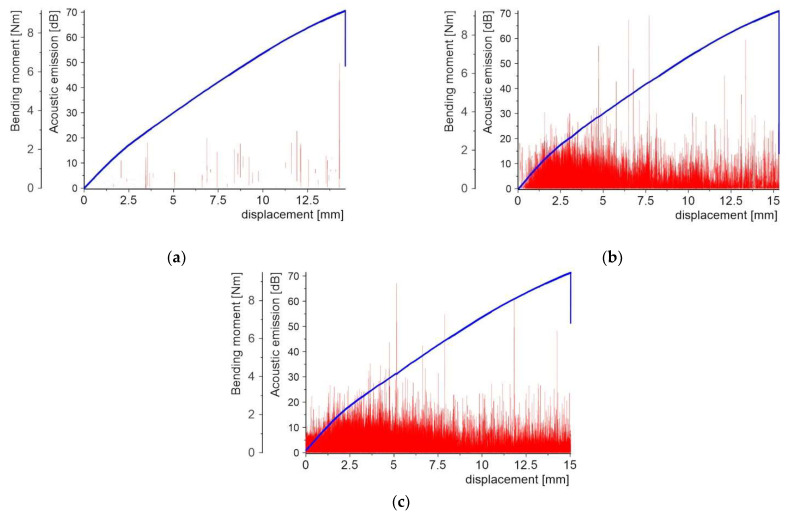
Bending moment and AE vs. displacement plots for samples with the PL made of crystalline silica powder: (**a**) sample 5_1; (**b**) sample 5_2; and (**c**) sample 5_3.

**Figure 13 molecules-27-01168-f013:**
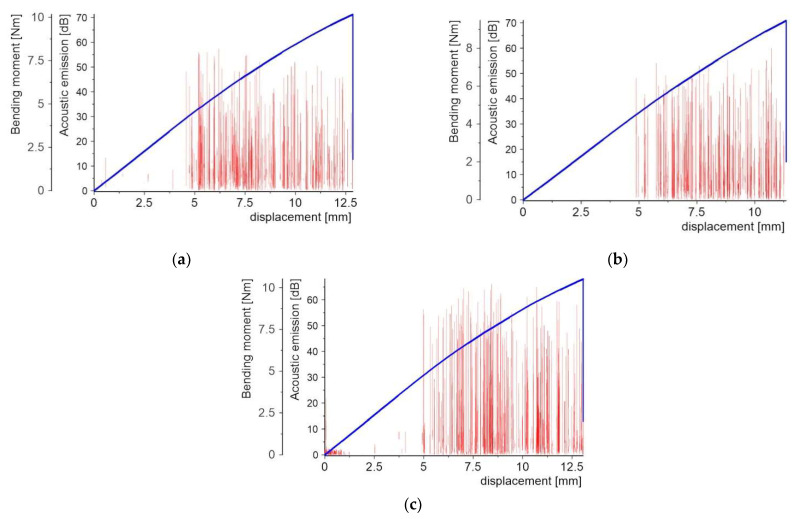
Bending moment and AE vs. displacement plots for specimens with microballoon layer: (**a**) sample 6_1; (**b**) sample 6_2; and (**c**) sample 6_3.

There are 10 to 13 smaller bending moment drops corresponding to the local damage events, which were confirmed by AE peaks before the complete failure of the samples. For samples 1_2 and 1_3, even before the first force drop, there was a noise level below 10 dB, which can be ignored in this case. In addition to peaks with significant values around 70 dB, peaks with values of 10–20 dB were present in the range from the first crack to the maximum load.

Contrary to the above, the specimens with the PL made of quartz sand were characterized by a very early cracking phenomenon. This occurs in the bending moment range from 1.77 to 2.09 Nm, which corresponds to a displacement range of 1.26 mm–1.49 mm. The stiffness of the specimen was found to significantly decrease after the appearance of the first crack. In Figure 9 several smaller bending moment drops are visible, but these no longer determine the reduction in stiffness. Thus, the first crack appears in the middle cross-section of the specimen and the damage zone is concentrated in the surrounding area. The other cracks appear successively from the central cross-section of the specimen to the supports. The bending moment drops coincide with the AE peaks, but their number is higher in comparison to the samples with the copper powder PL. The maximum values are at 70 dB level, but there is another zone of peaks at 30 dB, where their density is much higher. This level is responsible for the delamination of individual powder grains from the matrix. 

For samples with the Al_2_O_3_ powder PL (Figure 10), the first force drop appears at the bending moment equal to 2.57–2.91 Nm, which corresponds to the displacement of 2.36 mm–2.64 mm. The maximum values of the AE signals at 70 dB are related to the visible bending moment drops. There are a sparse number of signals at the tens of the dB level compared to the quartz powder.

In the case of the samples with the aluminum PL (Figure 11), the AE signal distri-bution was found to be completely different from the previous cases.

A few weak AE signals with the maximum values of order 30–40 dB, mainly be-fore the specimen final failure, are noticeable. The bending moment-displacement dia-grams are continuously increased, almost linearly, without any visible drop before the maxi-mum failure load. The absence of brittle fracture indicates that plastic defor-mation is occurring in the specimen, which was confirmed by microscopic observa-tions after specimen failure.

Application of the crystalline silica powder for the creation of the PL results in the absence of the local bending moment drops (Figure 12) during the 3-PB tests. This type of powder is a chemically similar material applied in the specimen with the PL made of quartz sand, but with a much smaller grain size and different shape. In contrast to samples with aluminum PL, there are several more AE signals with maximum values equal to 60–70 dB. There is also a zone with a significant number of signals in the range up to about 20 dB, which are probably related to the separation of individual grains from the matrix.

The last material used for the protective coating was the microballoon powder, for which the distribution of bending moments is different from the previous samples (Figure 13). There were no local decreases in bending moment found. The first AE sig-nals initiated for the bending moment were equal to about 4.6 Nm, which corresponds to a displacement of 5 mm and values at the level of 50–60 dB. The bending moment-displacement diagram becomes continuously nonlinear up to the final failure.

### 3.2. Numerical Simulation Results of the Cracking Process in the Central Part of the Specimen Subjected to 3-PB

The main objective of the numerical analysis was to determine how grain size and grain number influence microcrack initiation in the PLs and further their development to create the main vertical crack, which leads to the failure of the two-layered samples.

After computing, the models, the elastic, and the kinetic energy were analyzed. The latter is negligibly small up to the time of failure initiation. After microcrack creation, the elastic energy is partially released and transformed into kinetic energy. However, this stage of damage evolution was not analyzed in the present work. The present analysis is performed to the coalescence of vertical microcracks and the creation of dominant macrocracks through the PL thickness. 

Figure 14 presents the displacement maps just after the creation of the dominant vertical macrocrack in the PL. For model 1, it occurred at 85% (0.106 Nm) of the given load; for model 2, this occurred at 84% (0.105 Nm); and for model 3, it occurred at 66% (0.0825 Nm).

The presented results in Figure 14 lead to the conclusion that a decrease in the grain size accelerates the process of microcrack nucleation and further propagation. However, before it happens, damage initiates inside the PLs microstructure in different places that is not extraneously visible. Initiation of microcracks is responsible for the creation of a large number of AE signals during the deformation development. Figure 15 presents stress maps for the RVE models just before the vertical macrocrack creation by coalescence of microcracks. The differences between the three models are quite significant and explain why higher strengths were obtained for the larger grains:For model 1, microcracks in the matrix occur between grains, as in Figure 15a. However, there is still a visible matrix volume between the damage and the grain, hence stress redistribution occurs, which bypasses the crack and moves to the next grain.In model 2 (Figure 15b), the grains are more closely distributed to each other and, in addition to the damage in the matrix between the grains, damage also initiates at the grain–matrix interface.In model 3, where the distance between grains is the smallest, in the presented fragment in Figure 15c, the damage, in any case, appears at the grain–matrix contact.

Therefore, the load from the grain, due to the reduced surface area, causes a higher stress concentration, and thus a faster failure of the whole structure occurs. In model 3, it should also be noted that in the case of adjacent grains, the mentioned damage and grain separation from the matrix occurs only on one side; consequently, a situation of merging of close damage does not occur. This is because when one damage appears, the zone around it becomes relieved, which can be seen by the intense blue color.

The presented model, based on the continuum shell, also allows us to determine the effort level inside composite material (Figure 16). Its changes before and after the occurrence of a vertical crack can also be followed. In each case, there is a lack of symmetry in strain distributions along the vertical axis. The occurrence of the PLs causes strain relief in the lower zone. The highest effort occurs for powders with a diameter of 0.25 mm and 0.19 mm and is about 71%. The specimen with the PL built up of 0.15 mm grain size fails much earlier, hence the maximum effort is at 52%. So, in any case, after the PL fails, the composite could still carry the load.

The presented model is an idealized model; the distribution of grains is chaotic in reality as there are differences in shapes and sizes of grains, and the contact at the interface between the grains and the matrix is not continuous due to microvoids and other defects that appear just after the manufacturing process. However, this idealized model provides a glimpse into the interior of the structure. Future laboratory studies will focus on analyzing the effect of grain size and further development of the more advanced numerical model. 

## 4. Discussion of Results

Bending tests are commonly used in the analysis of composite materials under both static [36,37] and dynamic loads [38]. Especially for layered materials, these are a valuable source of information as they allow the observation of cracking and delamination phenomena. Section 3 presents detailed results for each specimen; however, establishing averaged values for each of the series are necessary for further analysis and discussion. 

### 4.1. Bending Moments and Failure Energy Estimated by Experiments

The maximum values of the bending moment, at which point the failure of the specimens occurred, are shown in Figure 17a; the highest value of 10.07 Nm was obtained for the specimens with the microballoon layer (6). The properties of the microballoon, with its bulk thickness of 1 mm in each case, allowed the formation of the thinnest PL to be about 0.3 mm. Thus, a relatively small amount of resin migrated from the prepreg to the PL. The slightly resin-free prepreg after curing had the greatest thickness of about 1.74 mm. Therefore, this is the explanation why the highest strength was obtained for the samples with the microballoon layer. 

A slightly lower value was obtained for the samples with aluminum powder (4), i.e., 9.87 Nm. For aluminum, the grain size is also at a low level, but the prepreg was desaturated from the resin to a greater extent as evidenced by microscopic measurements. In this case, a similar strength was obtained from a material that also experienced plastic deformation as a result of bending. 

The extreme case with the lowest strength was found to be the specimens with a layer of quartz sand (2), for which the averaged maximum moment value of 8.53 Nm was obtained. In this case, the largest size of grains of the PL was in the range of 0.1–0.4 mm. The PL thickness was equal to 1 mm, which corresponds to the thickness of the bulk layer consisting of solely grains, while the thickness of the PMC substrate was about 1.49 mm. The final strength of the whole system (substrate plus layer) depends on both powder grain size and the material of the PMC. 

Although the maximum bending moment was reached for the samples with the microballoon PL (6), the highest amount of energy at failure (42.46 J) must be supplied for the samples with the crystalline silica layer (5), as shown in Figure 17b. This results due to the total thickness of this protective system having a high thickness (2.48 mm) and the highest displacement at failure. For the other protective systems, the accumulated energy is in the range of 32.45 J–36.2 J.

### 4.2. Acoustic Emission (AE) Results Discussion

The AE experiments with the PMC can be characterized by several descriptors, such as:Peak amplitude;Peak frequency [39];Frequency centroid;AE energy [40];A cumulative number of hits [41].

Below, we discuss AE results for the description of the analyzed protective systems.

#### 4.2.1. Acoustic Emission Energy

The AE energy, *E_AE_*, can be estimated by integrating the transition voltage, $U_{i}^{}(t)$, of an acoustic event over a given time period. This energy in the time period from *t*_0_ to *t*_i_ can be expressed as:(2)$$E_{AE}=\int_{t_{0}}^{t_{i}}U_{i}^{2}(t)dt$$

Figure 18a shows the *E_AE_* for all analyzed protective systems. The highest value was obtained for quartz sand (2). It is notable that the quartz grains exhibited the largest size in the range of 0.1 mm–0.4 mm. Thus, the contact area between the grain and the matrix was relatively large, hence large acoustic energy was released during the separation of the epoxy resin and the quartz grains. 

The lowest energy value was achieved for the sample with aluminum powder (4). It is not possible to hypothesize that the cause was the material metal itself, since an almost 3.5 times higher emission energy was obtained for copper (1). The reasons should be seen both in the size of the grains, their shape, and the adhesion between the grains and the epoxy resin. The samples with a layer of aluminum powder were permanently deformed and no visible cracks appeared on the surface of the PL.

#### 4.2.2. Acoustic Emission Signals (AES)

The assessed AE energy is proportional to the number of counts of AE signals (Figure 18b). In this case, the highest number of AES (2082) was also achieved for samples with quartz sand PL (2). The lowest number (210) was achieved for samples with aluminum PL (4). The protective system with microballoon PL (6) exhibits a rather high value of AES counts (1043), similar to that with the copper (1063), but had the AE energy at a rather low level. It is 2.5 times lower compared to the system with copper PL (1). This indicates that the counts of AES or AE energy cannot be enough independently to evaluate the mechanical characteristics of the proposed PMC–PL systems.

#### 4.2.3. Acoustic Emission Peak Frequency (PF) and Frequency Centroid (FC) 

Another AE descriptor is the peak frequency (PF), i.e., the maximum magnitude in the power spectrum of a recorded acoustic event. The calculations were performed in Diadem software, where the recorded AE waveform was transformed using the fast Fourier transform (FFT) method.

The characteristic frequency spectra can be attributed to the density and stiffness of the materials used. Matrix cracking is characterized by lower frequencies, while fiber cracking emits higher frequencies due to its higher modulus. These three classes have been attributed to the three basic mechanisms of microscopic damage in composites: fiber cracking, matrix cracking, and interfacial cracking [32]. However, the authors in [40] conducted a study of composites using DIC and microscopic damage observations, and found that matrix cracking in CFRP composites can also lead to the emission of high-frequency signals. Therefore, the identification of high-frequency AE events associated with fiber cracking should be undertaken with caution.

The frequency centroid (FC) is also marked in Figure 19 and is a weighted average of frequencies expressed as:(3)$$\mathrm{FC}=\frac{\sum_{0}^{f\max}A(f)\cdot f}{\sum_{0}^{f\max}A(f)}$$

Compared to the PF, the FC is sensitive to the high-frequency content. The FC values for all sample types are close to each other and concentrated in the range from 422 kHz to 452 kHz, so the FC is not a good parameter to characterize the damage of the PL coating made of powder.

A much better parameter for damage description of the protective system is the PF. The obtained results lead to the conclusion that investigations of two-layered protective systems should be divided into two groups: The first group are powder materials used for the creation of the PLs, such as quartz sand and Al_2_O_3_, for which the PF value is close to FC;The second group obeys the rest of the applied powder materials, such as copper, aluminum, crystalline silica, and microballoon, where the PF value is almost half that of FC.

In the first case, the frequencies are close to 400 kHz, while in the second case, they are in the range of 200 kHz–260 kHz. The lowest PF value was achieved for the copper PL at 194 kHz, while the highest value was achieved for the quartz sand PL at 386 kHz.

## 5. Conclusions

The new manufacturing technology for the fabrication of a two-layered protective system made of multifunctional powder-based layers (PLs) joined to the PMC substrate in a single autoclave process was proposed in this study. 

The results indicate that the obtained new product is characterized by the absence of delamination of the layer, even at large deformations of the substrate. 

In summary, the following conclusions are drawn from the study:The damage and fracture characteristics of the proposed two-layered protective systems strongly depend on the powder used for the PLs and its grain size. For copper, quartz sand, and Al_2_O_3_ grains, there is a significant amount of local sharp drops in the bending moment-displacement plot. Such phenomenon was not observed for aluminum, crystalline silica, and microballoon grains used for the creation of the PLs.The best two-layered protective system, in terms of strength during 3-PB testing, was manufactured with the application of the aluminum grains PL. The acoustic emission signals, relative to the other systems, were at a low level equal to 30–40 dB maximum. In addition, the samples after the bending test were the only ones to undergo plastic deformation. The brittle fracture was very limited and the AE energy was minimal.In the case of the other two-layered protective systems containing the remaining five powder grains (copper, quartz sand, Al_2_O_3_, crystalline silica, and microballoon), the brittle microfracture was the most important mechanism responsible for damage growth leading to the final failure. The AE signals and AE energies were very high.For assessment of the damaging processes with different PLs built up of different powder grains by application of AE method, it was best to use the maximum amplitude instead of the centroid of the amplitude, which had almost the same value for the tested protective systems.The qualitative numerical analysis with the RVE model, selected from the central cross-section of the two-layered protective systems, explains the important role of the grain size in the multifunctional layer. When decreasing grain size diameter, the PL is degraded earlier. The microcracking occurs directly at the contact between the grain and the matrix.

## Figures and Tables

**Figure 1 molecules-27-01168-f001:**
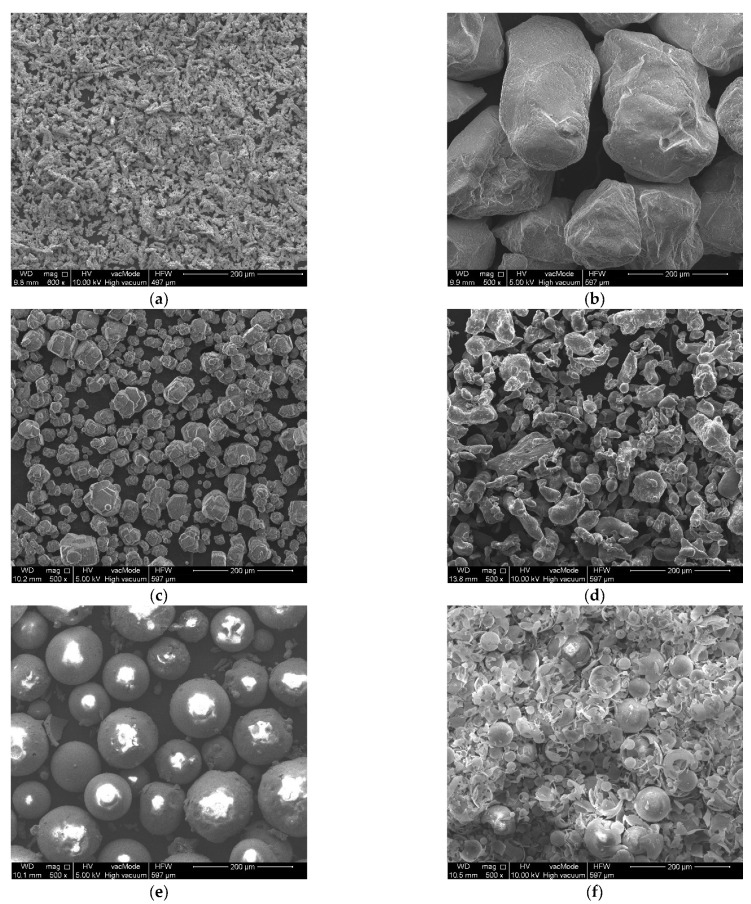
SEM microscope images for the applied powders: (**a**) copper; (**b**) quartz sand; (**c**) Al_2_O_3_; (**d**) aluminum; (**e**) crystalline silica; and (**f**) microballoon.

**Figure 2 molecules-27-01168-f002:**
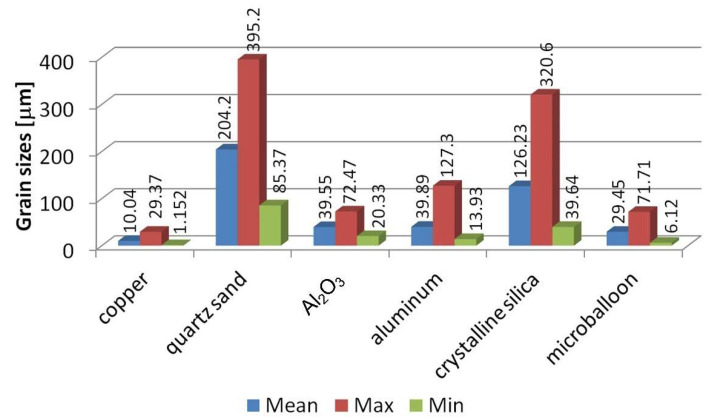
Grain sizes (μm).

**Figure 3 molecules-27-01168-f003:**
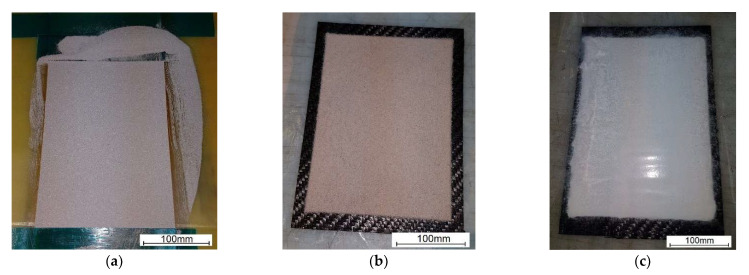
Fabrication technology of the PL: (**a**) use of a template; (**b**) remove of a template; (**c**) preparation for vacuum bagging technology.

**Figure 4 molecules-27-01168-f004:**
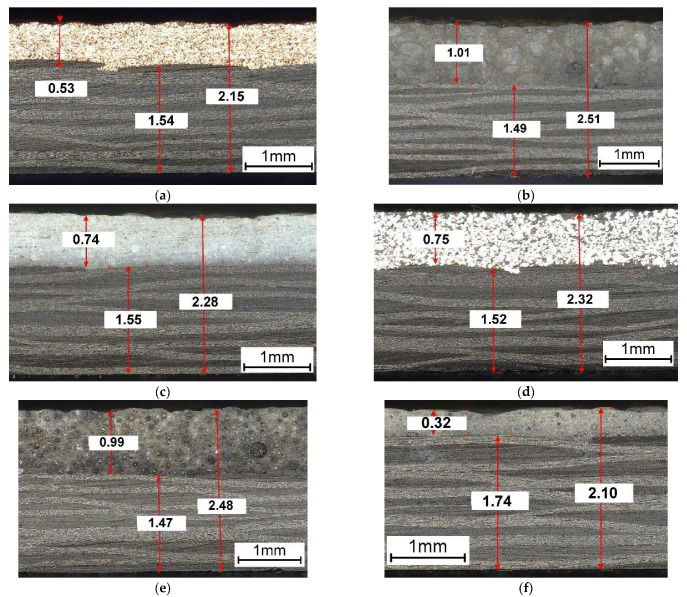
Internal structure of the two-layered composites consisting of the PMC substrates (**bottom**) with PLs (**top**) made of the following grains (dimensions are given in (mm)): (**a**) 1: copper; (**b**) 2: quartz sand; (**c**) 3: Al_2_O_3_; (**d**) 4: aluminum; (**e**) 5: crystalline silica; and (**f**) 6: microballoon.

**Figure 5 molecules-27-01168-f005:**
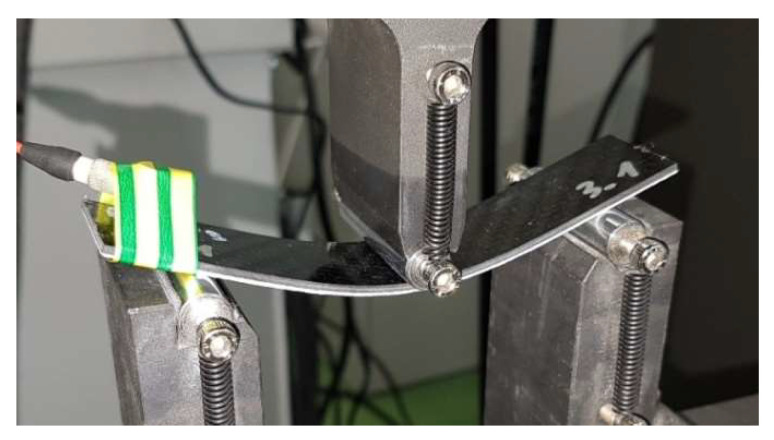
Displacement-controlled 3-PB deformation process of the sample with powder PL placed at the bottom.

**Figure 6 molecules-27-01168-f006:**
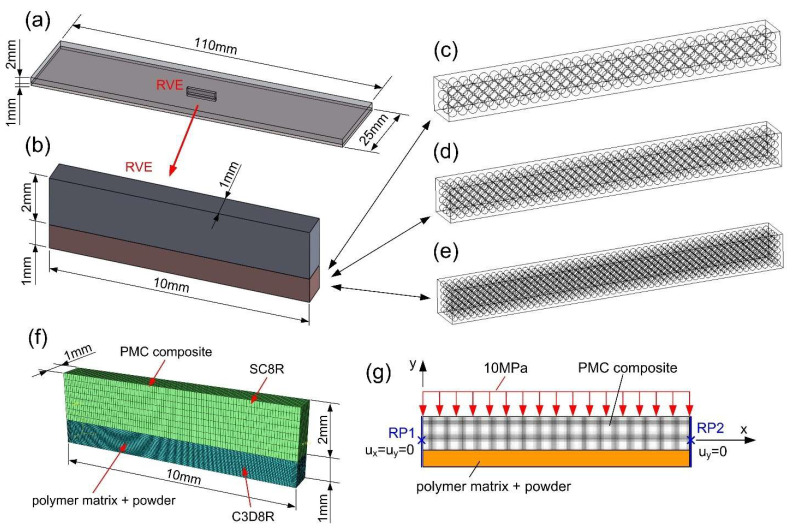
Numerical model: (**a**) global model; (**b**) RVE numerical models, having different numbers and sizes of powder grains, for the analysis of microcrack growth in the PLs; (**c**) model_1: 288 grains; (**d**) model_2: 672 grains; (**e**) model_3: 1300 grains; (**f**) RVE finite element structural mesh of the two-layered composite structure; and (**g**) boundary and loading conditions for numerical analysis of the RVE.

**Figure 7 molecules-27-01168-f007:**
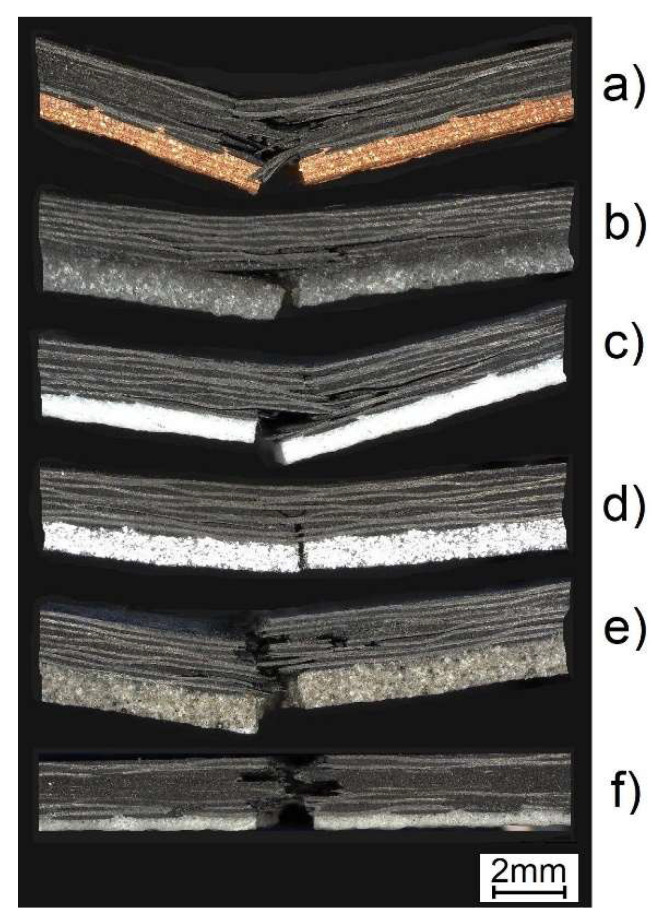
Failure modes of two-layered samples with PLs made of: (**a**) copper; (**b**) quartz sand; (**c**) Al_2_O_3_; (**d**) aluminum; (**e**) crystalline silica; and (**f**) microballoon powders.

**Figure 14 molecules-27-01168-f014:**
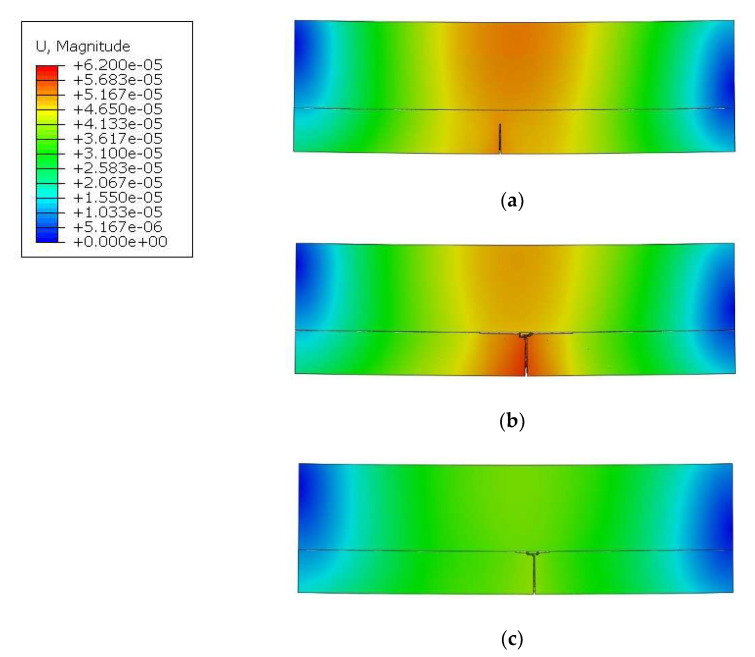
Displacement maps of the RVE models: (**a**) model_1 (fi = 0.25 mm); (**b**) model_2 (fi = 0.19 mm); and (**c**) model_3 (fi = 0.15 mm).

**Figure 15 molecules-27-01168-f015:**
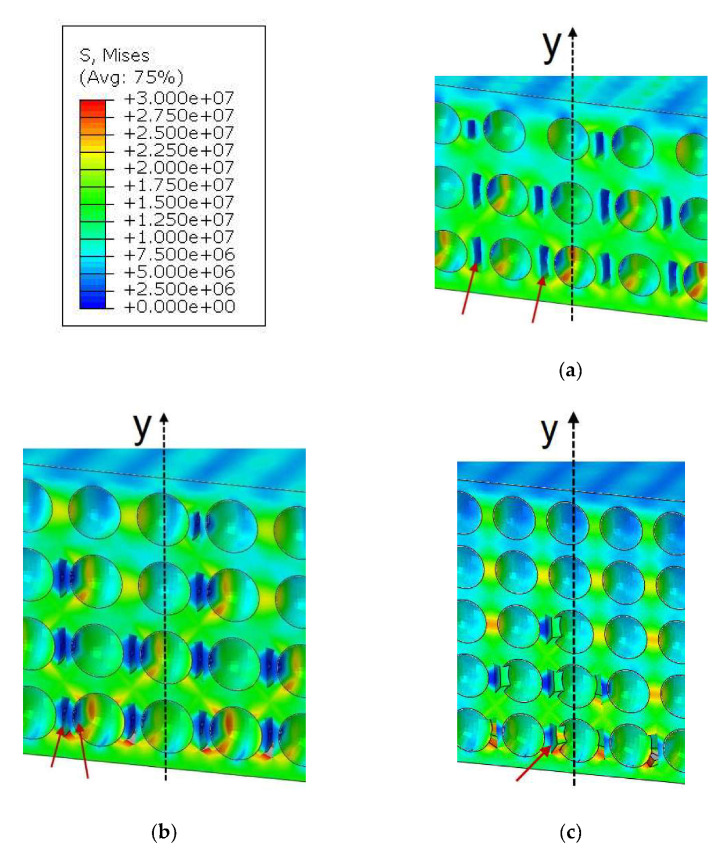
Tensile damage of the matrix in the PL by vertical microcracking: (**a**) model_1 (fi = 0.25 mm); (**b**) model_2 (fi = 0.19 mm); and (**c**) model_3 (fi = 0.15 mm).

**Figure 16 molecules-27-01168-f016:**
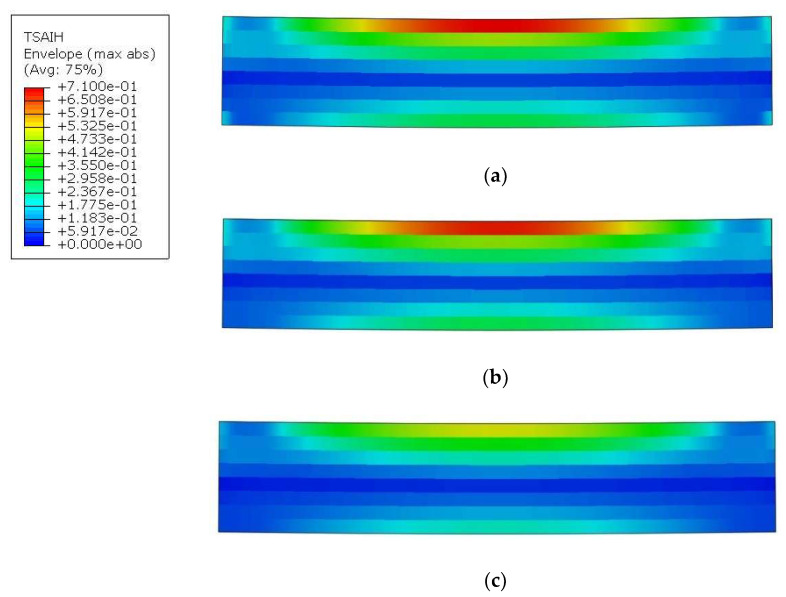
Tsai–Hill criterion maps: (**a**) model_1 (fi = 0.25 mm); (**b**) model_2 (fi = 0.19 mm); and (**c**) model_3 (fi = 0.15 mm).

**Figure 17 molecules-27-01168-f017:**
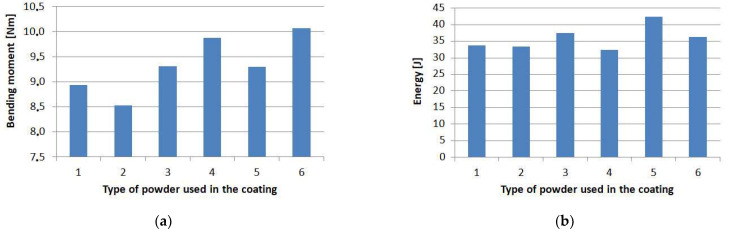
Results of experiments: (**a**) averaged maximum bending moment values; and (**b**) averaged energy values.

**Figure 18 molecules-27-01168-f018:**
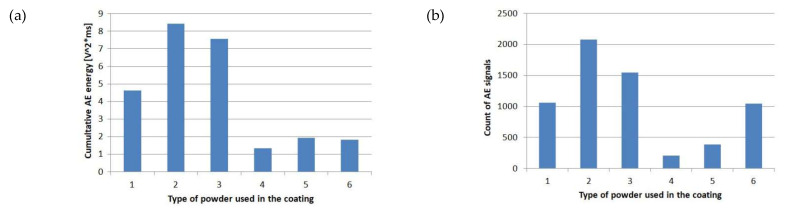
AE results: (**a**) acoustic emission energy *E_AE_*; and (**b**) number of acoustic emission events.

**Figure 19 molecules-27-01168-f019:**
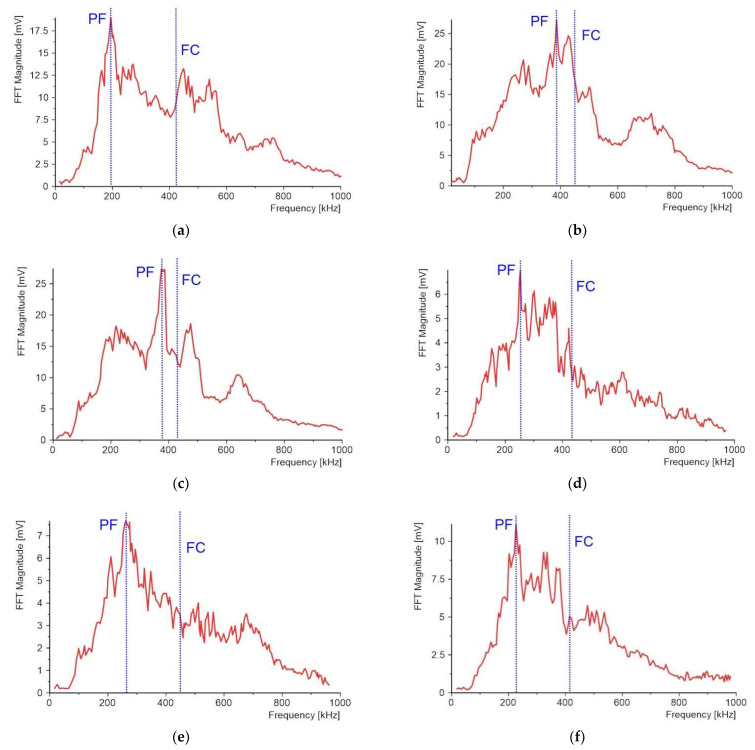
Frequency spectra: PF and FC: (**a**) copper powder; (**b**) quartz sand powder; (**c**) Al2O3 powder; (**d**) aluminum powder; (**e**) crystalline silica powder; (**f**) microballoon powder.

**Table 1 molecules-27-01168-t001:** Properties of the Kordcarbon prepreg.

Tensile strength (0°)	657.3 MPa [30]
Tensile modulus	51.1 GPa [30]
Poisson ratio ν_12_	0.11 [30]
Shear strength	118.1 MPa [30]
Shear modulus	3.167 MPa [30]
Compression strength (0°)	479.6 MPa [30]

**Table 2 molecules-27-01168-t002:** Properties of the epoxy matrix.

Tensile strength	20.7 MPa [31]
Modulus of elasticity	523 MPa [31]

## Data Availability

Not applicable.

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
