# Peer review of "Technological and Strength Aspects of Layers Made of Different Powders Laminated on a Polymer Matrix Composite Substrate"

_molecules, 2022, doi:10.3390/molecules27041168_

Round 1
Reviewer 1 Report
The authors investigate a new polymer matrix composite aiming at thermal barrier coatings.
The following issues arise when reading the paper:
- In the introduction, no clear statement of the research goal and of the state-of-the-art appears.
- The paper is not organized. The methodical parts apper in various parts of the text. It looks, like the goal is to compare the experimental and numerical results of 3-point bending of the PMC. Therefore, the paper must be rearranged, by grouping all the experimental materials and methods, and also numerical methods into Section 2. Then, 3. Experimental and Numerical Results ahould appear. After that, the discussion can be understandable. Acoustic emission results (5.2) shpuld appear in the results section 3.
- Protective layer (PL) contraction appear only in the abstract, it is also needed in the introduction, as with PMC.
- Microbaloon elemental/phase content must appear in Section 2 -Materials and methods.
- Figure 3 lacks the scale. Also, the numbers in the figure need units. Is it mocrometers?
- The first paragraph and Fig. 4 in Section 3 still belong to the Materials and Methods.
- The Conclusions are too wordy. some of the text should be moved into the Discussion section. It is not clear, which coating is better.
- The phrase like "the most optimal two-layered protective system" is unacceptable. The choice is either optimal according to a selected criterium (and certain constraints), or not optimal.
- No papers from year 2021 are cited. This raises the concern that the authors are not aware of the recent research in the field.
Author Response
Reviewer 1
The authors investigate a new polymer matrix composite aiming at thermal barrier coatings.
The following issues arise when reading the paper:
- In the introduction, no clear statement of the research goal and of the state-of-the-art appears.
Answer: The introduction has been expanded to include new literature and the text has been reorganized.
- The paper is not organized. The methodical parts apper in various parts of the text. It looks, like the goal is to compare the experimental and numerical results of 3-point bending of the PMC. Therefore, the paper must be rearranged, by grouping all the experimental materials and methods, and also numerical methods into Section 2. Then, 3. Experimental and Numerical Results ahould appear. After that, the discussion can be understandable. Acoustic emission results (5.2) shpuld appear in the results section 3.
Answer: In-depth changes have been made to each of the chapters. Section 5.2 was renumbered to 4.2 and remained in the "Discussion" section because it deals with the post-processing analysis of the results using the appropriate descriptors.The raw results from the acoustic emission are presented in Section 3.
- Protective layer (PL) contraction appear only in the abstract, it is also needed in the introduction, as with PMC.
Answer: explanation of the acronym was added.
- Microbaloon elemental/phase content must appear in Section 2 -Materials and methods.
Answer: The chemical composition for the microballoon was added.
- Figure 3 lacks the scale. Also, the numbers in the figure need units. Is it mocrometers?
Answer: Scale has been added, dimensions are in millimeters.
- The first paragraph and Fig. 4 in Section 3 still belong to the Materials and Methods.
Answer: The text at the beginning of Section 3 and Figure 4 were moved to Section 2.
- The Conclusions are too wordy. some of the text should be moved into the Discussion section. It is not clear, which coating is better.
Answer: Conclusions were shortened; some text was moved to Discussion. Changes were made in the text to emphasize that the best coating is made of aluminum powder.
- The phrase like "the most optimal two-layered protective system" is unacceptable. The choice is either optimal according to a selected criterium (and certain constraints), or not optimal.
Answer: Changes were made in the text, this is the bending strength criterion.
- No papers from year 2021 are cited. This raises the concern that the authors are not aware of the recent research in the field.
Answer: New articles [6] and [8] of 2021 were added.

Reviewer 2 Report
The manuscript entitled “Technological and strength aspects of layers made of different powders laminated on a polymer matrix composite substrate for further creation of thermal barrier coatings” explored a description of the new technology for producing external or internal layers made of different powders mixture with epoxy resin, which can perform various functions as a protection against impact, erosion, or elevated temperatures. The research is interesting and worth to be investigated, however, major revision is need.
- The language should be significantly improved, for instance, remove those words: ‘The paper’, ‘They’, etc.
- In the keywords, please use the full name but not abbreviation.
- The introduction could be extended. Please include some information of biomass-based composites. A few suggested articles as follows: a) Preparation and properties of cellulose nanocomposite fabrics with in situ generated silver nanoparticles by bioreduction method; b) Cellulose nanocomposites: Fabrication and biomedical applications; c) Bacterial cellulose/glycolic acid/glycerol composite membrane as a system to deliver glycolic acid for anti-aging treatment
- The references cite format should be corrected entire manuscript, for instance, ‘In [9-14]’ (Line 65), which is not right.
- The data in Table 1 should be converted into figure. The particle distribution graph should be given.
- In Figure 3, what’s the unit in the figures?
- The same situation in Figure 5. There is no scale bar.
- Too much figures in the manuscript. I suggested to combined some of them into one, or put some into the supporting information.
Author Response
Reviewer 2
The manuscript entitled “Technological and strength aspects of layers made of different powders laminated on a polymer matrix composite substrate for further creation of thermal barrier coatings” explored a description of the new technology for producing external or internal layers made of different powders mixture with epoxy resin, which can perform various functions as a protection against impact, erosion, or elevated temperatures. The research is interesting and worth to be investigated, however, major revision is need.
- The language should be significantly improved, for instance, remove those words: ‘The paper’, ‘They’, etc.
Answer: Language changes in the text are highlighted in yellow.
- In the keywords, please use the full name but not abbreviation.
Answer: full names have been added.
- The introduction could be extended. Please include some information of biomass-based composites. A few suggested articles as follows: a) Preparation and properties of cellulose nanocomposite fabrics with in situ generated silver nanoparticles by bioreduction method; b) Cellulose nanocomposites: Fabrication and biomedical applications; c) Bacterial cellulose/glycolic acid/glycerol composite membrane as a system to deliver glycolic acid for anti-aging treatment
Answer: Three new literatures were added as suggested by the reviewer.
- The references cite format should be corrected entire manuscript, for instance, ‘In [9-14]’ (Line 65), which is not right.
Answer: The authors use the program "Mendeley", the style was imported from a CSL file for the journal "Molecules".
- The data in Table 1 should be converted into figure. The particle distribution graph should be given.
Answer: The table was converted into a drawing.
- In Figure 3, what’s the unit in the figures?
Answer: dimensions are given in [mm].
- The same situation in Figure 5. There is no scale bar.
Answer: scale bar was added.
- Too much figures in the manuscript. I suggested to combined some of them into one, or put some into the supporting information.

Round 2
Reviewer 1 Report
The authors introduced significant changes into the paper text. However, some issues still arise when reading the paper.
- The coating the authors investigate is not a thermal barrier coating (TBC), it is just a sublayer which probably can be used for TBC applications. Therefore, the title must be corrected accodingly. The same in the Introduction, lines 70-75. Actually, your conclusion says nothing about TBC - because the paper is not about TBC!
- The abstract is too wordy. Lines 9 to 17 must be packed into one sentence. The idea of the numerical modelling coupled with the experimental investigations must appear as a solid methodology, not like an addition.
- In my previous comments, I specifically asked to introduce the goal of the research and the state of the art. This was not corrected. Instead, the authors added the methodological description in lines 88-90 and 94-97. Please distinct the Methods and Introduction.
- Figure 3 lacks the scale marker.
- Figures 4 and 7 now have the markers, but the red color cannot be read normally, please change the color or baskground. Do you need it red? Is it a warning?
- The section Materials and Methods must be split into subsections - for experimental and numerical methods, and separately, materials and their characterization methods.
Author Response
Reviewer 1
The authors introduced significant changes into the paper text. However, some issues still arise when reading the paper.
- The coating the authors investigate is not a thermal barrier coating (TBC), it is just a sublayer which probably can be used for TBC applications. Therefore, the title must be corrected accodingly. The same in the Introduction, lines 70-75. Actually, your conclusion says nothing about TBC - because the paper is not about TBC!
Answer: Title was changed. The text in the range of lines 70 - 75 is very important to the authors because we are currently conducting attempts to produce TBC layers on the interlayers presented in this paper.
- The abstract is too wordy. Lines 9 to 17 must be packed into one sentence. The idea of the numerical modelling coupled with the experimental investigations must appear as a solid methodology, not like an addition.
Answer: Lines 9 through 17 were combined into one sentence.
- In my previous comments, I specifically asked to introduce the goal of the research and the state of the art. This was not corrected. Instead, the authors added the methodological description in lines 88-90 and 94-97. Please distinct the Methods and Introduction.
Answer: The final section of the Introduction was rewritten.
- Figure 3 lacks the scale marker.
Answer: Scale was added.
- Figures 4 and 7 now have the markers, but the red color cannot be read normally, please change the color or baskground. Do you need it red? Is it a warning?
Answer: Background was introduced, color was changed to black.
- The section Materials and Methods must be split into subsections - for experimental and numerical methods, and separately, materials and their characterization methods.
Answer: Chapter 2, was divided into two subsections.

Reviewer 2 Report
It's ready to be published.
Author Response
We improved the manuscript according to the first round of the review process.